# Mapping the intermolecular interaction universe through self-supervised learning on molecular crystals

## Abstract

Molecular interactions fundamentally influence all aspects of chemistry and biology. Prevailing machine learning approaches emphasize the modeling of molecules in isolation or at best provide limited modeling of molecular interactions, typically restricted to protein-ligand and protein-protein interactions. Here, we present how to use molecular crystals to define the MOLINTERACTDB dataset that contains valuable biochemical knowledge, which can be captured by large self-supervised pre-trained models. MOLINTERACTDB incorporates 344,858 molecular crystal structure entries from the Cambridge Structural Database. We formulate entries in the MOLINTERACTDB dataset as radial patches of flexible size and at varying positions in the crystal to represent intermolecular interactions across crystal structures. We characterize a variety of interactions highlighted across 6 million patches. Leveraging MOLINTERACTDB, we develop INTERACTNN, a self-supervised SE(3)-equivariant 3D message passing network. We show that INTERACTNN captures the latent knowledge of chemical elements as well as intermolecular interaction types at a scale not directly accessible to human scientists. To demonstrate its potential, we fine-tuned INTERACTNN to predict the binding affinity between proteins and ligands, producing results comparable with state-of-the-art models.

## 1  Introduction

Intermolecular interactions between molecules play a central role in understanding and predicting chemical phenomena [7, 17, 27, 65]. In drug discovery, intermolecular interactions between the ligand and target are key factors for the selectivity and specificity of the drug [52, 68, 3, 46, 34, 19, 71]. While these interactions are important for chemists, the exploration of intermolecular interactions in machine learning is limited. Many state-of-the-art models in molecular property prediction train on molecular datasets featuring molecules in isolation, for PCQM4Mv2 [39], QM9 [43], and ZINC [18]. In contrast, ML models for molecular interactions are restricted to protein-ligand and protein-protein interaction (PPI) structures, leaving the broader field of intermolecular interactions largely untouched. Given the fundamental role of intermolecular interactions, it is important to consider a broader variety of these interactions to improve the generalizability of ML models.

An experimental data modality that captures intermolecular interactions is a crystal structure [40], which records the 3D coordinates of the atoms in the crystal. In molecular crystals, molecules are bound together by intermolecular interactions in an infinitely repeating lattice. To represent this periodic structure, crystal structures are expressed as a unit cell—the smallest repeating unit of the crystal. There are large datasets of crystal structures including the Cambridge Structural Database

Submitted to NeurIPS 2023 AI for Science Workshop.

[13] (CSD) featuring 1,222,711 entries of which 344,858 are of organic molecular crystals that satisfied our search criteria. Unlike the discrete molecules found in many molecular property datasets, representing unit cells to capture intermolecular interactions for ML presents unique challenges.

**Present work.** We present MOLINTERACTDB, a dataset created from the CSD that captures intermolecular interactions from unit cells of entries in the CSD. Each entry in the MOLINTERACTDB is a radial patch which includes the 3D coordinates and atomic identities of intermolecular interacting molecular fragments. A key contribution of MOLINTERACTDB is the expansion of intermolecular examples available for ML models. By leveraging the CSD, we extend beyond molecular interactions limited to protein-ligand and protein-protein complexes, and present more intermolecular interactions that are chemically relevant for ML. In addition to dataset creation, we also developed the INTER-ACTNN model which is trained with self-supervised objectives to learn an informative latent space of patches. Probing the latent representation space reveals its ability to learn chemical types of interactions, and elemental differences. Finally, we show that INTERACTNN can be fine-tuned to predict binding affinity of protein-ligand interactions and achieves comparable results to state-of-the-art.

## 2   Related work

**Machine learning for molecules.** In the field of molecular machine learning, there are various studies ranging from property prediction to generation. Molecules can be represented either as 1D strings, such as SMILES [61] and SELFIES [25], and are typically trained using language models [72, 60]. Alternatively, 2D and 3D molecular structures can be represented as graphs [44, 67, 69, 11, 48, 64, 32] and trained using graph neural networks (GNNs). These models can predict molecular properties [29, 41, 32] and help design new molecules [34, 42, 70, 28, 21, 35, 62].

**Geometric deep learning for molecular prediction and design.** Molecules can adopt multiple 3D configurations, known as conformers, which are not represented in 1D or 2D forms. Additionally, 3D geometric information significantly influences the properties and functions of a molecule. Consequently, several geometric deep learning models incorporate 3D coordinates for molecular property prediction [47, 9, 31, 58]. Given the scarcity of labeled 3D molecular data, self-supervised formulations for pre-training on 3D molecular structures have been developed. Notable models include GraphMVP [30], GNS-TAT [66], and 3D InfoMax [51]. Among them, GNS-TAT [66] demonstrates that pre-training by denoising 3D structures towards equilibrium can enhance performance in downstream tasks. Subsequently, these models are fine-tuned on smaller 3D molecular datasets with labeled molecular properties. Progress has also been made in constructing equivariant models. These ensure that when certain symmetry operations or transformations are applied to the input, equivalent transformations are reflected in the output. This is crucial for maintaining the consistency of output predictions with SE(3)-symmetry operations, which include translations and rotations [45, 14].

**Machine learning for molecular interactions.** Molecular interactions underpin virtually all processes within living organisms. Several models have been developed to predict molecular interactions, including binding affinity prediction [37, 63, 38, 26, 37], binding site prediction [36, 24, 20, 22], and PPI prediction [7, 54, 8]. The field of molecular interactions is expanding with emerging areas of interest such as the design of molecular glues to stabilize PPIs [4] and the modulation of PPIs to target the undruggable proteome [33]. However, the scarcity of data, primarily due to the challenges in capturing 3D molecular data of interacting biological compounds, has curtailed the widespread application of ML in these nascent fields. Recognizing the need to understand intermolecular interactions across stages of drug design and development and across therapeutic modalities, we harness large datasets of molecular crystals to advance the modeling of intermolecular interactions.

## 3   Creating MOLINTERACTDB dataset

In this section we outline how we curate the CSD to capture examples of intermolecular interactions from molecular crystals. We start by defining intermolecular patches (Sec. 3.1) and proceed by outlining the curation of MOLINTERACTDB (Sec. 3.2).

## 3.1 Overview of Cambridge Structural Database (CSD)

The CSD [13] contains all known crystal structures of small-molecule organic and metal-organic crystal structures. These structures are experimentally determined with X-ray or neutron crystallography. As of 2023 there are over 1.25 million crystal structures in the CSD, of which under half of these structures are classified as organic. A review from Taylor and Wood highlights the contributions of the CSD in researching molecular geometries, interactions, and assemblies [55].

**Curating molecular cystals from the CSD.** Querying and accessing of crystal structure data was done with the CSD Python API. Each entry of the CSD describes a crystal structure stored as a unit cell, the smallest component that represents the repeating crystal structure, and the metadata including publications associated with the entry, experimental details, and the chemical formula. Additionally, the CSD computationally assigns bonds and bond types between atoms to every entry. An example of data available for an entry in the CSD is shown in Figure 1. We filtered CSD v2022.3.0 for all entries that satisfied all of the following criteria: organic, not polymeric, has 3D coordinates, no disorder, no errors, no metals, had only one SMILES string describing the crystal entry (in other words, each crystal is comprised of only one chemical compound). This filtered the CSD dataset from 1,222,711 entries to 344,858.

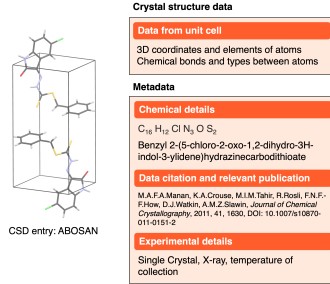

Figure 1: Illustrative example of the CSD entry for ABOSAN.

## 3.2 Creating intermolecular patches in MOLINTERACTDB using molecular crystals

To represent intermolecular interactions we define intermolecular patches as entries in MOLINTERACTDB. Each radial patch is centered between two molecules to capture the geometric orientations of non-bonding interactions between two molecules. This approach captures diverse types of intermolecular interactions, including hydrogen bonding, Van der Waals interactions, aromatic interactions. Here we do not directly model the periodic unit cell, as we focus on recording intermolecular interactions. Radial patches have been shown to be useful in related fields of modeling protein surfaces [7, 54, 8, 53, 2], where patches are defined on the surface of a protein to reduce large protein surfaces to a fingerprint. Our approach differs to the use of patches for modeling of protein surfaces—which only feature one molecule—instead our patches capture interactions between molecules.

**Definition 3.1** (**Intermolecular Patch**). An intermolecular patch $G^{(ij)}$ is a graph with geometric 3D coordinate attributes that is comprised of molecular fragments of intermolecularly interacting molecules $i$ and $j$, here denoted as $M^{(i)}$ and $M^{(j)}$, respectively. Intermolecular interactions are all non-bonding interactions between $M^{(i)}$ and $M^{(j)}$; this includes hydrogen bonding, dipole-dipole interactions, Van der Waals interactions, and aromatic-aromatic interactions. Molecular fragments in the patch are all atoms in the molecules that are within a radius $r$ from the weighted center, $c^{(ij)} = 1/(2|M^{(i)}||M^{(j)}|) \left( |M^{(j)}| \sum_{k \in M^{(i)}} \mathbf{p}_k + |M^{(i)}| \sum_{k \in M^{(j)}} \mathbf{p}_k \right)$, where $\mathbf{p}_k$ is the atomic coordinates of the molecules. The nodes and edges of the $G^{(ij)} = \left( V_{G^{(ij)}}, E_{G^{(ij)}} \right)$ are:

- **Nodes:** $V_{G^{(ij)}} = \left( V^{(i)}, V^{(j)} \right)$, where $V^{(i)}, V^{(j)}$ are atoms in $M^{(i)}$ and $M^{(j)}$ that are within radius $r$ to the center $c^{(ij)}$. We denote arbitrary nodes in $V_{G^{(ij)}}$ with $a$ and $b$.
- **Edges:** $E_{G^{(ij)}} = \left( E^{(ij)}, E^{(i)}, E^{(j)} \right)$ are comprised of:
  - **Intermolecular edges:** $E^{(ij)}$ connect atoms in $V^{(i)}$ with atoms in $V^{(j)}$ that are positioned within distance $d_{\text{inter}}$ of each other.
  - **Intramolecular edges:** $E^{(i)}$ are edges between atoms in $V^{(i)}$, and $E^{(j)}$ are edges between nodes in $V^{(j)}$ that are within distance $d_{\text{intra}}$.

We also refer to neighbouring nodes $b \in \mathcal{N}_a^{(t)}$ of node $a$ in a patch, where $t \in \{\text{inter}, \text{intra}\}$ are the intermolecular and intramolecular edge neighbours. For intermolecular neighbours, we refer to the

edges $E^{(ij)}$. For intramolecular neighbours, if $a \in V^{(i)}$ we refer to the edges $E^{(i)}$, otherwise if $a \in V^{(j)}$ we refer to the edges $E^{(j)}$.

The MOLINTERACTDB dataset, $\mathcal{D} = \{G^{(i_k j_k)} \mid k = 1, \ldots, N\}$, is comprised of patches $G^{(ij)}$ constructed from CSD entries. For our purposes of learning intermolecular interactions we sampled many patches to represent all examples of intermolecular interactions in a unit cell (Figure 2). Given an entry of the CSD, we iterate through each unique, valid conformer $M^{(i)}$ in the unit cell using the CSD Python API. For each conformer $M^{(i)}$ we iterate through all neighbouring peripheral conformers $M^{(j)}$ of this molecule given by the unit cell that are within $d_{\text{inter}}$ to an atom in $M^{(i)}$. A patch $G^{(ij)}$ is constructed from all atoms in $M^{(i)}$ and $M^{(j)}$ that are within radius $r$ to the weighted center $c^{(ij)}$. This extraction of patches from a unit cell will yield some patches that are equal up to permutation.

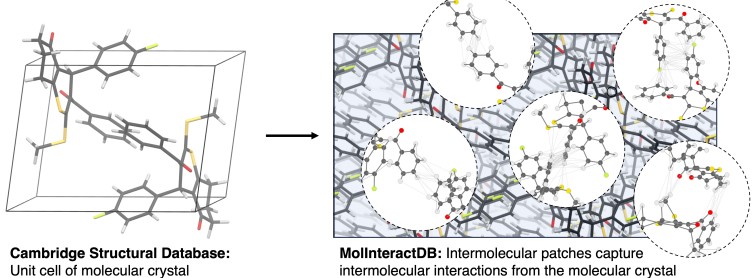

**Cambridge Structural Database:**
Unit cell of molecular crystal

**MolInteractDB:** Intermolecular patches capture
intermolecular interactions from the molecular crystal

Figure 2: Intermolecular molecular patches from MOLINTERACTDB. CSD entry ABIGAV is shown.

We set $r = 8$ Å $d_{\text{inter}} = 4$ Å and $d_{\text{intra}} = 2$ Å. After iterating through all 344,858 CSD entries that satisfied our CSD filters, this constructs 6,059,368 patches in $\mathcal{D}$. The choice of radius, intermolecular and intramolecular edge distance cutoff for the patch will influence the number of patches created. A radius $r$ that is too small would break basic chemical motifs, which would lead to insufficient chemical context for interactions in the patch. Intramolecular edge cutoffs $d_{\text{intra}}$ that are too short would also disregard longer chemical bonds, and intermolecular edge cutoffs $d_{\text{inter}}$ that are too short would limit the number of patch examples. Our choice of cutoffs aim to provide sufficient chemical context. We summarise statistics of the patches in MOLINTERACTDB in Table 1.

Table 1: Properties of 6,059,368 patches in MOLINTERACTDB with $r = 8$ Å, $d_{\text{inter}} = 4$ Å, and $d_{\text{intra}} = 2$ Å.

| Graph features | | | | | Chemical features | |
|---|---|---|---|---|---|---|
| Feature | Mean | SD | Min | Max | Element | % Distribution |
| # Nodes | 67.8 | 24.3 | 4 | 424 | Carbon | 44.6 |
| # Intermolecular edges | 34.1 | 34.5 | 1 | 8,547 | Hydrogen | 42.4 |
| # Intramolecular edges | 89.5 | 36.6 | 2 | 2,859 | Oxygen | 6.1 |
| Intermolecular node degree | 1.3 | 0.2 | 0.0 | 22.9 | Nitrogen | 3.9 |
| Intramolecular node degree | 0.5 | 0.4 | 0.1 | 11.45 | Fluorine | 0.8 |

## 4 INTERACTNN model and its compelling use cases

Next we outline the INTERACTNN model that uses MOLINTERACTDB for self-supervised pre-training. We provide details for how we probe the learned latent space of the INTERACTNN to explore the space of chemical interactions (Sec. 4.1) and show how INTERACTNN can be fine-tuned for protein-ligand binding prediction (Sec. 4.2).

## 4.1 Overview of INTERACTNN model

INTERACTNN uses a SE(3)-equivariant 3D message passing network on intermolecular patches to learn representations that are informative of the intermolecular interaction between molecules.

**Problem (Self-Supervised Pre-Training For intermolecular Patches).** *Given is an unlabeled pre-training dataset of intermolecular patches, $\mathcal{D} = \{G^{(i_k j_k)} \mid k = 1, \ldots, N\}$, and a target dataset of labeled intermolecular patches $\mathcal{S} = \{(G_{\text{target}}^{(i_k j_k)}, y_k) \mid k = 1, \ldots, M\}$, where $M << N$. Our goal is to pre-train a model $\mathcal{F}$ on $\mathcal{D}$ such that it generates representations $\mathbf{z}_k = \mathcal{F}(G^{(i_k j_k)})$ for every intermolecular patch $G^{(i_k j_k)}$ that are chemically informative, and $\mathcal{F}$ can also be fine-tuned on $\mathcal{S}$ to predict $y_k$ for every $G_{\text{target}}^{(i_k j_k)}$.*

**Atom-level representation learning.** Here we outline the SE(3)-equivariant 3D message passing network for INTERACTNN on the nodes of the intermolecular patch $G^{(i,j)}$. Several rotational equivariant neural networks have been introduced for modeling molecules [49, 23, 31, 1]. We build on the E(3)-equivariant neural network layers presented by Tensor-Field Networks implemented in e3nn [10] and DiffDock [3]. Message passing for the intermolecular edges and intramolecular edges are done separately, but the message passing framework for the two edge types is the same.

The feature vectors $\mathbf{h}_a$ of nodes $a$ in $G^{(i,j)}$ are geometric objects that comprise a direct sum of irreducible representations of the O(3) symmetry group. The feature vectors $\mathbf{h}_a^{(\lambda,p)}$ are indexed with $\lambda, p$, where $\lambda = 0, 1, 2, \ldots$ is a non-negative integer denoting the rotation order and $p \in \{\text{o}, \text{e}\}$ indicates odd or even parity, which together index the irreducible representations (irreps) of O(3). There are also multiple features in $\mathbf{h}_a$ which have the same irrep. In our model, we set $\lambda_{\max} = 1$ for $\mathbf{h}_a$, and we denote the number of scalar (0e) and pseudoscalar (0o) irrep features in $\mathbf{h}_a$ with ns, and the number of vector (1o) and pseudovector (1e) irrep features in $\mathbf{h}_a$ with nv.

First, the element type of node $a$ is embedded with a normal distribution and trainable weights to a vector with feature configuration ns $\times$ 0e. The edge length between the coordinates of node $a$ and neighbouring node $b$ is also embedded with Gaussian smearing to a vector comprised of ns $\times$ 0e, then the Gaussian embedding vector is passed through a 2-layer MLP projector to output a feature vector $\mathbf{e}_{ab}$ with feature configuration ns $\times$ 0e.

There are $L$ layers of message passing between nodes. At each layer $l$, the node updates for node $a$ in the intermolecular patch $G^{(i,j)}$ are given by:

$$\mathbf{h}_a \leftarrow \mathbf{h}_a \underset{t \in \{\text{inter,intra}\}}{\oplus} \text{BN}^{(t)} \left( \frac{1}{\left|\mathcal{N}_a^{(t)}\right|} \sum_{b \in \mathcal{N}_a^{(t)}} Y^{(\lambda)}(\hat{r}_{ab}) \otimes_{\psi_{ab}} \mathbf{h}_b \right) \text{ with } \psi_{ab} = \Psi^{(t)}\left(\mathbf{e}_{ab}, \mathbf{h}_a^{0\text{e}}, \mathbf{h}_b^{0\text{e}}\right), \quad (1)$$

where node $b$ are the neighbours of node $a$ in $G^{(i,j)}$ given by intermolecular or intramolecular edges denoted with $t$. The message is computed with tensor products between the spherical harmonic projection with rotation order $\lambda = 2$ of the unit bond direction vector, $Y^{(\lambda)}(\hat{r}_{ab})$, and the irreps of the feature vector of the neighbour $\mathbf{h}_b$. This is a weighted tensor product and the weights are given by a 2-layer MLP, $\Psi^{(t)}$, based on the 0e features of the nodes $\mathbf{h}_a$ and $\mathbf{h}_b$ and the edge features $\mathbf{e}_{ab}$. After each layer $l$ of message passing, $\mathbf{h}_a$ is filtered down to irreps with $\lambda_{\max} = 1$. After $L$ layers the final irreps configuration of $\mathbf{h}_a$ is ns $\times 0e + \text{nv} \times 1o + \text{nv} \times 1e + \text{ns} \times 0o$ and the embedding of node $a$, $\mathbf{h}_a$ is a $d_{\text{node}}$-dimension vector.

**Intermolecular patch-level representation learning.** For a patch-level embedding of the nodes a convolution is done between all nodes in a molecule and the unweighted center $c^{(i)}$ of the nodes $V^{(i)}$. This is repeated for nodes $V^{(j)}$ in the patch $G^{(ij)}$. The edge distance from node $a$ to $c^{(i)}$ is also embedded with Gaussian smearing and passsed through a 2-layer MLP projector to output a feature vector $\mathbf{e}_{ac^{(i)}}$ with feature configuration ns $\times$ 0e as:

$$\mathbf{h}_{c^{(i)}} = \text{BN} \left( \frac{1}{\left|V^{(i)}\right|} \sum_{a \in V^{(i)}} Y^{(\lambda)}(\hat{r}_{a,c^{(i)}}) \otimes_{\gamma_{ac^{(i)}}} \mathbf{h}_a \right) \text{ with } \gamma_{ac^{(i)}} = \Gamma\left(\mathbf{e}_{ac^{(i)}}, \mathbf{h}_a^{0e}\right). \quad (2)$$

This is a weighted tensor product and the weights are given by a 2-layer MLP projector, $\Gamma$, based on the 0e features of the nodes $\mathbf{h}_a$ and the edge features $\mathbf{e}_{ac^{(i)}}$. The embedding of the intermolecular

patch $G^{(i_k j_k)}$ is given by $\mathbf{z}_k = [h^{0e}_{c^{(i_k)}} || h^{0e}_{c^{(j_k)}}]$, the concatenation of the scalars from embedding molecule $i_k$ and $j_k$, which is a $d_{\text{patch}}$-dimension vector.

**Self-supervised training with denoising.** Node-level denoising as an objective function has been useful for pre-training on 3D coordinate molecular datasets from DFT generated molecules to prevent over-smoothing of GNNs [12], and it has proven that it is related to learning a force field of per-atom forces [66, 6]. In addition, denoising is linked to score-matching which has also been popular in training generative models [16, 3]. Thus, this motivates the application of denoising as an objective for self-supervised training on MOLINTERACTDB.

Given a patch $G^{(i,j)} \in \mathcal{D}$, $\tilde{G}^{(i,j)}$ is a perturbed patch created by adding i.i.d. Gaussian noise to the atomic positions, $\mathbf{p}_a$ of each node $a \in V_{G^{(ij)}}$. That is, for each node $a \in V_{\tilde{G}^{(ij)}}$ the atomic position $\mathbf{p}'_a = \mathbf{p}_a + \boldsymbol{\delta}_a$, where $\epsilon_a \sim \mathcal{N}(0, \sigma I_3)$, $\sigma = 0.5$ and $\boldsymbol{\delta}_a = \min(\epsilon_a, \mathbf{1})$. The objective is to predict $\{\boldsymbol{\delta}_1, \ldots, \boldsymbol{\delta}_{|V_{\tilde{G}^{(ij)}}|}\}$ given $\tilde{G}^{(i,j)}$. The model $\mathcal{F}_{\text{denoise}}$ is trained to minimise the loss $\mathcal{L}$:

$$\mathcal{L} = \sqrt{\frac{1}{N} \sum_{G^{(i,j)} \in \mathcal{D}} \| \mathcal{F}_{\text{denoise}}(\tilde{G}^{(i,j)}) - (\boldsymbol{\delta}_1, \ldots, \boldsymbol{\delta}_{|V_{\tilde{G}^{(ij)}}|}) \|^2} \tag{3}$$

We add a denoising layer to $\mathcal{F}$ for $\mathcal{F}_{\text{denoise}}$ to predict the noise applied for each node from $\tilde{G}^{(i,j)}$. This final layer of the message passing on $\tilde{G}^{(i,j)}$ takes as input the node-level embeddings $\mathbf{h}_a$ and is the same message passing framework as outlined in Eq. (1). However, the output irreps are restricted to $1 \times 1o + 1 \times 1e$. To convert this to 3D coordinates, the $1 \times 1o$ and $1 \times 1e$ are summed element-wise to produce a vector in $\mathbb{R}^3$ and the prediction is clamped to the maximum noise applied which is 1 Å.

### 4.2 Implementation and use cases

**Implementation.** The model was trained with the denoising objective with a batch size of 64 on MOLINTERACTDB for 48 hours on 48GB RTX 8000 GPUs. The model hyper-parameters were set to ns = 32, nv = 16, $L = 6$, and $lr = 1 \times 10^{-3}$.

**Probing latent representation space.** We investigate whether the self-supervised training of the model resulted in a chemically meaningful latent space by characterizing the patch-level and node-level embedding spaces of MOLINTERACTDB. To determine the chemical labels of patches, we convert the molecular fragments within a patch into RDKit molecules and sourced labels from RDKit. Nodes are labeled based on their atomic elements and further categorized by examining the elements they were bonded to, as well as the bond types.

**Modeling protein-ligand binding.** In this use case, we use the PDBbind v2020 dataset [59], which is a curated subset of the Protein DataBank (PDB) with the structure of bound ligands to proteins, and the associated binding affinity. The task is: given the protein-ligand structure, predict the binding affinity. We use the pocket-ligand substructures of the protein-ligand structure given by PDBbind, where amino acids in the pocket are all amino acids with any atom within 6 Å to the ligand. Given the pocket-ligand, we construct a graph with the same features as a patch where the two molecules in the patch are the pocket and the ligand, and intermolecular edges are defined as edges between the pocket and the ligand. Note that we do not restrict the size of the pocket-ligand patch to a radial cutoff. The pocket-ligand graph is passed through the pre-trained INTERACTNN which gives a patch-level embedding that is passed through a 3-layer MLP predictor to output a binding affinity prediction clamped to between 0 and 15. The INTERACTNN is fully fine-tuned on the pocket-ligand structures to minimize the root mean squared error between predicted and experimental binding affinity.

## 5 Experiments

### 5.1 Use case: Probing the latent space of chemical interactions

**Setup.** Given the pre-trained INTERACTNN we embedded all the nodes and patches and visualized a 2D UMAP for each set of nodes and patches. For patches we label the types of intermolecular interactions at the interface between the two molecules in the patch. If any of the intermolecular

interactions are between two atoms in an aromatic system, the patch is labeled as aromatic. Otherwise, if any of the intermolecular interactions are between two atoms where one is a hydrogen bond donor and another is a hydrogen bond acceptor, the patch is labeled as hydrogen bonding. Other interactions, such as dipole-dipole, and Van der Waals interactions are not labeled. For node embeddings, we labelled each node with the atomic element. For the most common elements, carbon and hydrogen, we explored with further granularity by considering the elements the nodes are bonded to and bond types. To test the statistical significance of chemical clusters we use the Kolmogorov-Smirnov (KS) test to compare randomly sampled pairwise distances of $d$-dimensional embeddings compared to pairwise distances sampled within $d$-dimensional embeddings of the same chemical label.

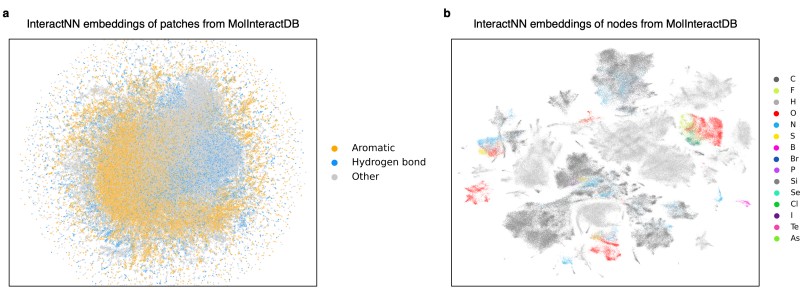

Figure 3: 2D UMAP plots of INTERACTNN embeddings of (**a**) 300,000 randomly sampled patches from MOLINTERACTDB. Each dot is a patch and they are labeled by the type of intermolecular interactions present in the patch. (**b**) 300,000 randomly sampled nodes from patches from MOLIN-TERACTDB. Each dot is a node and they are labeled by element of the node.

**Results.** The INTERACTNN learns an overall embedding space for patches as well as nodes in every patch and we find that embeddings are meaningfully localized based on various chemical properties. In Figure 3, we use a 2D UMAP to visualize the embedding of 300,000 randomly sampled patches from MOLINTERACTDB. Labeling of the UMAP with the chemical type of intermolecular interaction as aromatic groups interacting with aromatic groups, or hydrogen bond donor and hydrogen bond acceptor shows INTERACTNN learns a chemically enriched latent space in a self-supervised manner. We also see statistically significant differences with $p$-value $< 0.001$ for the pairwise distance of embeddings labeled as hydrogen or aromatic against all patch-level embeddings.

Visualization of the embedding space of 300,000 sampled nodes of patches from MOLINTERACTDB in Figure 4 highlights that INTERACTNN has learnt differences in atomic environments in a self-supervised manner. In Figure 4a, we see in the embedding space that the INTERACTNN has differentiated between the elements. Isolating the most common elements, hydrogen, and carbon, pairwise distance of node embeddings within these elements are statistically significantly different to pairwise distances of all node embeddings ($p$-value $< 0.001$). We also show that the embeddings of carbon and hydrogen nodes can be stratified further by the bonding environment. Remarkably, without any prior knowledge of bond types, Figure 4e shows that INTERACTNN embeds the aromatic carbons in a separate region to the aliphatic carbons (single bonded carbons).

## 5.2 Use case: Protein-ligand binding affinity prediction

A sequence-based split of 60% from Atom3D [56] is used to train and test the model. We compare our protein-ligand binding affinity prediction with state-of-the-art models trained and tested under the same dataset split. Performance is determined by minimizing the root mean squared error between predicted and actual binding affinity, and by maximizing Pearson and Spearman correlation coefficients between the predicted and actual binding affinity. Results in Table 2 show that the performance of INTERACTNN is comparable to state-of-the-art models across all metrics. We also show that the absence of pre-training for INTERACTNN results in a decay in performance.

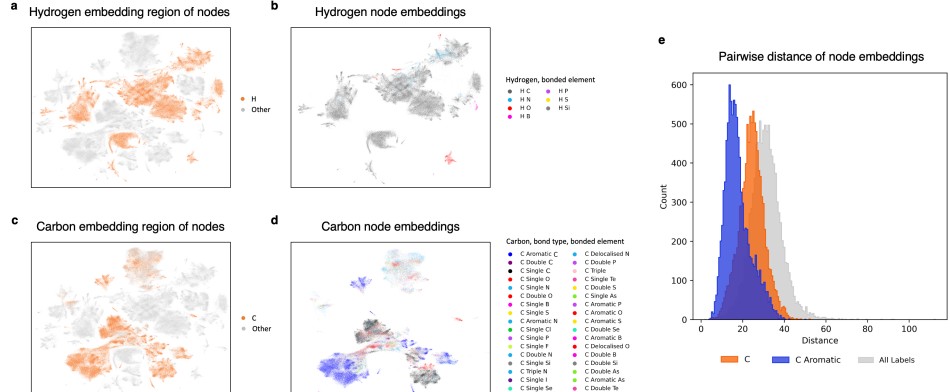

Figure 4: 2D UMAP plots of INTERACTNN embeddings of 300,000 randomly sampled nodes from patches in MOLINTERACTDB. (**a**) Highlighting the hydrogen nodes against other elements. (**b**) Each dot is a hydrogen node and they are labeled by the element that hydrogen is bonded to. (**c**) Highlighting the carbon nodes against other elements. (**d**) Each dot is a carbon node and they are labeled by the type of bond, and element that carbon is bonded to. (**e**) Pairwise distance of node embeddings with a given label. All distributions are statistically significantly different from each other (two-sided non-parametric KS test; $p$-value < 0.001).

Table 2: Results on protein-ligand binding affinity task with 60% sequence identity split. The top two results are highlighted as **1st** and 2nd. We report the benchmark metrics provided by ProNet [57].

| Method | RMSE ↓ | Pearson ↑ | Spearman ↑ |
|---|---|---|---|
| Atom3D [56] | 1.408 | 0.743 | 0.743 |
| ProtTrans [5] | 1.641 | 0.595 | 0.588 |
| MaSIF [7] | 1.426 | 0.709 | 0.701 |
| IEConv [15] | 1.473 | 0.667 | 0.675 |
| Holoprot [50] | 1.365 | 0.749 | 0.742 |
| ProNet [57] | **1.343** | **0.765** | **0.761** |
| INTERACTNN | 1.355 | 0.748 | 0.746 |
| INTERACTNN no pre-training | 1.415 | 0.719 | 0.717 |

## 6  Conclusion

Intermolecular interactions are essential to chemical properties and diverse functions of biological systems. In this work, we introduce a MOLINTERACTDB dataset that leverages large molecular crystal databases to extract examples of intermolecular interactions between molecular fragments in the form of intermolecular patches. We explore the diversity of this dataset and train a INTERACTNN model on MOLINTERACTDB in a self-supervised manner. We show that the learned latent space of INTERACTNN is informative for capturing nuances between hydrogen bonding and aromatic interactions. The model can also distinguish between chemical elements. Finally, we fine-tune the model for protein-ligand binding affinity prediction and achieve results comparable to state-of-the-art models. In the future, we will adapt INTERACTNN for fine-tuning on other molecular interaction tasks, including protein-protein interactions, and explore the model's ability for few-shot prompting and zero-shot learning.

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
