# OpenReview forum: "Mapping the intermolecular interaction universe through self-supervised learning on molecular crystals"
_NeurIPS.cc/2023/Workshop/AI4Science — NeurIPS2023-AI4Science Poster_

### Meta-Review · Area_Chair_YJeW · 2023-10-26

**Recommendation:** Accept (Poster)
**Confidence:** 2

**Metareview:**

The manuscript introduces a novel molecular interaction dataset, MolInteractDB, derived from molecular crystal structures, offering insights beyond conventional protein-ligand and protein-protein interactions. The authors further develop InteractNN, a self-supervised SE(3)-equivariant 3D message passing network, trained on this dataset, which demonstrates promise when fine-tuned for protein-ligand binding affinity prediction.

No reviews were submitted for this paper, making the decision challenging. I looked into this paper and the comments are as follows.

Good points:
1. The paper presents a comprehensive dataset, MolInteractDB, comprising 344,858 molecular crystal structure entries, enhancing the scope of molecular interaction study beyond prevalent paradigms.

2. Pre-trained on the MolInteractDB dataset, the model, when fine-tuned to predict protein-ligand binding affinities, demonstrates competitive performance, aligning it with existing state-of-the-art models.

**Concerns**:
1. **Writing Clarity**: The clarity in some sections of the manuscript could be enhanced for better comprehension.

2. **Downstream Applications**: exploring and demonstrating the model's efficacy on a broader range of downstream tasks could enhance this paper.

Considering the contributions made by introducing the MolInteractDB dataset and developing the InteractNN model, along with competitive fine-tuned predictions, the paper holds good value. Based on these merits, I support the **acceptance** of the paper. Future iterations would benefit from addressing the concerns raised, primarily enhancing the writing clarity and expanding on the downstream applications of the model.